# The Effects of Different Relative Loads in Weight Training on Acceleration and Acceleration from Flying Starts

**DOI:** 10.3390/sports10100148

**Published:** 2022-09-27

**Authors:** Jøran Ersdal Fossmo, Roland van den Tillaar

**Affiliations:** Department of Sport Sciences and Physical Education, Nord University, 7600 Levanger, Norway

**Keywords:** 1-RM, strength training, effect size, acceleration, flying start

## Abstract

The purpose of this review was to examine how different relative loads in weight training can improve acceleration over 10 m from a standing or flying start. A systematic review of the literature was undertaken using the following databases: PubMed, MedLine, Google Scholar, and SPORTDiscus. Studies were eligible if they met the following criteria: (1) participants were at least 15 years or older and healthy and injury free, (2) the study included at least one exercise for the lower body with a strength training frequency of at least once a week and included a training period of at least four weeks, and (3) interventions with clear pre- and post-test results on 10 m sprint or 10 m flying start are stated. Non-English-language articles were excluded. Percent change and between-group effect size (ES) were calculated to compare the effects of different training interventions. Forty-nine studies met the inclusion criteria. The results were categorized into four groups: (1) explosive weight training with light loads at 30–60% of 1-RM, (2) explosive weight training with moderate loads at 60–85% of 1-RM, (3) maximal weight training at 85–100% of 1-RM, and (4) hypertrophy training at 60–85% of 1-RM. At 10 m, all methods of weight training demonstrated improvements, and maximal weight training demonstrated the highest results with a large ES, while other approaches varied from very small to moderate ES. Weight training showed little progression with a significantly lower effect on flying start across all training methods, except for one group that trained power cleans (hypertrophy) where progress was large. To improve acceleration over the first 10 m, this review demonstrated maximal weight training as the preferred training method. For athletes with a pre-existing high level of strength, it could be more appropriate to use explosive training with light loads or a combination of the two. To a lesser extent, acceleration from a flying start could be improved using both training methods as well.

## 1. Introduction

Sprinting is an important ability in several sports. Sprinting varies from being highly important in certain sports, such as the 100 m sprint where sprint velocity is fundamental to performance, to more versatile sports such as football, rugby, and American football, where sprinting is one of several important athletic abilities [1]. Overall, performance in a 100 m sprint can be divided into three phases: acceleration, maximal speed, and deceleration [2,3]. Mero, et al. [2] described the three phases as an acceleration phase beginning at zero, followed by an acceleration of bodyweight by increasing stride length and step frequency until reaching a maximal stride length and step frequency. This is normally achieved within 30–50 m before ending with a deceleration phase towards the end of the 100 m sprint. At the same time, the length of the three phases will vary between individual athletes based on training and genetic profile [4]. Acceleration is described as an increase in speed over a certain distance, and maximal speed is the highest speed a person can reach over a certain distance [5]. An improvement in maximal speed is strongly related to performance during a 100 m sprint, while maximal power is relevant to sprinting performance in a 100 m race. The connection to performance is even stronger over shorter distances [6].

In team sports like baseball, football, and American football, a significant difference in speed has been measured from the top to the lower ranks [7,8,9]. In American college football, a significant difference in 40-yard sprint time was registered between players in the upper division compared to divisions 2 and 3 [8]. This trend was also noted among elite football/soccer players who had a higher level of performance over 10 m compared to non-elite/junior players [9]. The same trend was again noted among elite baseball players over 10 yards compared to those in the lower divisions [7]. This highlights that speed is an important factor that can separate semi-professional athletes from professional and elite athletes. This is due to speed playing an important role in several key situations in each sport. For example, football players with greater speed will be able to position themselves faster into critical situations during competition to win the ball and create goal scoring opportunities [10,11].

In many sports, the acceleration phase is considered more important than maximal speed [12]. For instance, in sports such as squash, basketball, and tennis, athletes never reach maximal speed, as they are focused on the repetition of short sprints across small, limited areas [12]. In football and rugby, most of the sprints are shorter than 20 m [13,14]. When assessing the running times of male sprint finalists and semi-finalists over 100 m during the 1988 Olympics, the athletes reached 45%, 84%, 93%, and 97% of their maximal speed at 10, 20, 30, and 40 m, respectively [15], which demonstrated that when starting at zero and accelerating up to 10 m, a person will only ever achieve a relatively small portion of his or her maximal speed. This means that the near-maximum speed is not relevant to sports that contain sprints around 10 m, such as tennis and squash [16]. In some sports like football and rugby, however, this could be relevant, as both sports have moving starts [17,18,19].

### 1.1. Factors Determining Sprinting Performance

It is generally accepted that weight training can improve running speed [20]. This is demonstrated by the strong correlation among maximal strength, relative strength, and the development of maximal power [21,22,23], which means that sprinting performance is heavily influenced by maximal strength, relative strength, and the rate of force development (RFD) in the lower body [20].

These factors are influenced by muscle fiber composition, the nervous system’s ability to coordinate movement, muscle fiber length, muscle group cross-sectional area, and spring sensitivity in the musculotendinous system [24,25]. A large cross-sectional area is important to develop maximal force, and a correlation has been found between a larger muscular cross-sectional area around the medial area of the knee and an improved performance time in the 100 m sprint [26]. A high proportion of type 2 muscle fibers is necessary to develop force quickly, which, in turn, affects sprinting performance [27,28]. Longer muscle fibers contract quicker, and there is a correlation between long muscle fibers and an improved performance over 100 m [29,30,31]. Spring sensitivity in the musculotendinous system is also an important performance factor in the stretch–shortening cycle that occurs during plyometric activity, which refers to the contact time between running steps in a sprint. A strong correlation between the stretch–shortening cycle and performance during 30 m and 100 m races has been demonstrated [32]. These abilities are again affected by neural factors, such as the recruitment of motor units, firing frequency, and synchronization between motor units and intermuscular coordination [24,33]. The ability to activate all motor units quickly is important in order to develop a large amount of force rapidly, while signals from motoneurons to the motor units will determine how fast and large force development to the muscles will be [33]. Synchronization between motor units is considered to improve coordination between muscles, where the antagonist, synergist, and agonist receive the correct activation to develop force as quickly and significantly as possible in relation to the movement being carried out [33].

### 1.2. Weight Training to Improve Speed

To improve speed through weight training, the aim is to influence one or several of the important factors within relevant musculature that play a part in speed development. The Hill’s curve [34] shows an inverse relationship between speed and force in a muscle; a muscle produces a large amount of force over a low shortening speed, while at a high shortening speed the force is low. The Hill’s curve also demonstrates the ability to develop maximal power by detailing how the largest product of force x speed also produces the greatest maximal power [33]. Maximal power can be viewed as the desire to complete the greatest amount of work over the shortest amount of time and is, as mentioned, correlated with speed [22]. In order to improve athletes’ Hill curve, it is therefore important to increase the cross-sectional area of the muscles, achieve longer muscle length, influence muscle fiber composition, and improve their ability to activate muscle mass quickly and maximally [33].

A training program with maximal weight training over a sufficient length of time, with a resistance of at least 80% will achieve hypertrophy and an increase in maximal strength. This can therefore, in theory, provide an increase in force across all shortening speeds [35,36,37]. Maximal weight training can also increase the activation of type 2 muscle fibers, provided they are required to develop near-maximal force, according to Henneman’s size principle [38], causing them to be recruited and trained more often. This also leads to a hypertrophy of type 2 fibers, thus creating a larger cross-sectional area across the muscle group and leading to the development of higher levels of force [39,40]. Training for muscle growth leads to hypertrophy of muscle fibers in the cross-sectional area and can therefore also theoretically improve development of force across all shortening speeds [37].

Another type of weight training to improve speed is explosive weight training with light loads, also called power training, which is used to focus on activation of all high-threshold motor units, increase firing frequency, and improve intramuscular coordination. This leads to an increase in RFD and improvement in maximal power [25]. The intention behind lifting with maximal speed throughout the movement, regardless of high or low resistance, is to make the activity more sports specific in relation to whether the athlete is sprinting, jumping, or throwing with maximal mobilization [25]. The combination of explosive weight training with light loads and heavy weight training can improve muscular performance across high shortening speeds and muscular performance in low shortening speeds in relation to the force-velocity curve [41]. weight training can also cause muscle fibers to increase in length, shorten faster, and therefore create a faster RFD [27,29]. Plyometric exercises intended to improve the stretch-shortening cycle have been shown to influence muscle fiber force potential and lead to faster contraction and an increase in maximal power because of more optimal spring sensitivity [37]. As weight training can provide greater force development and faster RFD, contact time is reduced between steps, step frequency is higher, and, in return, acceleration is better during a sprint [42,43].

Considering all the factors mentioned within the muscular and nervous system, it is reasonable to conclude that muscle strength is an essential factor during a sprint, particularly in the acceleration phase [20]. Therefore, the primary aim of this review is to investigate how explosive weight training with loads at 30–60% of one repetition maximum (1-RM), explosive weight training with loads at 60–85% of 1-RM, maximal weight training at 85–100% of 1-RM, and hypertrophy training at 60–85% of 1-RM can improve acceleration over a 10 m distance. The secondary aim is to explore what weight training provide the best effects on a flying start, which refers to a particular starting form in sports in which the contestants are already in full motion when they cross the starting line.

## 2. Materials and Methods

### 2.1. Literature Search

The literature review was carried out in February 2022 by using the online databases PubMed, MedLine, Google Scholar, and SPORTDiscus. The following keywords were used in various combinations: «10 m sprint», «sprint», «acceleration», «flying time», «flying start» «strength», «resistance», «velocity», «training», «explosive», «light», «heavy», and «performance». Combinations of these keywords were used to identify the most relevant materials for review. Study selection was determined after a process of reading the title, abstract, and study to identify suitability in terms of the inclusion criteria. Further studies were also discovered by reading the reference lists from the papers chosen in the initial selection and through reading meta-analyses that have considered similar variables as in this review.

### 2.2. Inclusion and Exclusion Criteria

Inclusion criteria included participants having an age of consent of at least 15 years in addition to being healthy and injury free. The weight training interventions chosen by the authors had to include at least one exercise for the lower body. The studies also had to have a weight training frequency of at least once a week and include a training period of at least four weeks. Finally, the studies had to have been written in English, and access to a full-text document had to be available. All the studies selected for the review had included at least one of the mentioned weight training regimes and included the 10 m sprint as a pre- and post-assessment. Some studies also carried out a form of testing with moving start, which was also included in the review. Figure 1 shows the complete searching process through a PRISMA flowchart.

After a review of the online databases and after searching through reference lists, a total of 42 articles that fit the inclusion and exclusion criteria were selected. These studies are placed in Table 1, Table 2, Table 3 and Table 4 according to each of the training methods selected, showing information about the study participants, training intervention, percentage progression from pre-test to post-test, and effect size (ES) of the training intervention. The percentage change and ES were calculated in Microsoft Excel and were used to determine which approach provided the best training effect of all the weight training programs. The ES was calculated using Cohen’s d formula, which indicates a low effect in the range of 0.2–0.5, moderate at 0.5–0.8, large at 0.8–1.2, very large at 1.2–2, and huge at >2 [44,45].

## 3. Results

Of the 42 studies, there were 71 experimental groups including a total of 822 participants. Of all the participants, 292 were athletes from football, 124 from rugby, 59 from futsal (a smaller, indoor version of football), 34 from handball, 28 from rowing, 20 from basketball, 11 athletes at college level and 10 from mixed martial arts. Of the remaining participants, 73 were defined as resistance-trained athletes, 146 as physically active, and 16 as relatively weak/untrained. The average age of participants was 21.0 years, and each experimental group contained on average 11.6 participants. Sixty-four of the experimental groups were composed of only men, five of only women, one group where the gender of the subjects was not specified and only one group was mixed gender. Of the experimental groups, 20 performed maximal weight training (85–100% of 1-RM), 20 performed explosive weight training with moderate loads (60–85% of 1-RM), 20 performed explosive weight training with light loads (30–60% of 1-RM), and 11 focused on training for hypertrophy (60–85% of 1-RM). Further information about each of the experimental groups and training programs is included in Table 1, Table 2, Table 3 and Table 4.

### 3.1. Maximal Weight Training (85–100% of 1-RM)

On average, maximal weight training demonstrated the largest percentage progression and largest increase in ES over 10 m compared to all other training, with an average of 4.23% progress, placing the ES at 1.48 (Table 5). The percentage change varied from −3.33% [53] to 11.68% [51] (Figure 2). The ES varied from no ES to huge ES (Figure 3). Half squats and squats were the most utilized training form in maximal weight training. Of the 20 groups, 18 groups included a training frequency of two sessions a week, and the majority included a training period of eight weeks. The study with the highest percentage progression trained several exercises where half squats with four sets at a resistance of 85% of 1-RM during each session where amongst one of the exercises [51]. All groups except for two [53,55] experienced an improvement in their 10-m sprint at the end of the training period. Out of the 20 groups, six groups experience a huge ES (>2), three groups a very large ES (>1.2), four groups a large ES (>0.8), and the remaining seven a very low (>0.01) to moderate (>0.5) ES (Figure 2). Of the 13 groups that experienced a large ES (>0.8), 12 of them did share a common trait of being from ball-based sports. Out of the 20 groups, five measured a form of flying start between 10 and 40 m. On average, maximal weight training provided a percentage progression on flying start of 0.91% (Figure 4), while the average ES was low (0.30) (Figure 5). Percentage progression varied from 0% [58] to 1.54% [58]. Two of the groups experienced a moderate ES (>0.5), one group a low ES (>0.2), and the remaining groups a very low ES (>0.01) (Figure 5). A common trait among the two experimental groups that achieved the upper end of the moderate ESs was that they both were from the same study [56].

### 3.2. Explosive Weight Training with Moderate Loads (60–85% of 1-RM)

On average, explosive weight with moderate load training demonstrated a percentage progression in the 10-m sprint of 2.69%, while average ES was the second lowest out of all the training methods at 0.34 (Table 5). The percentage change (Figure 2) varied from −4.89% [66] to 10.81% [61]. The study with the highest percentage progression trained half-squats at six repetitions for four sets at 80% of 1-RM [61]. The ES in the groups varied from no ES to very large ES (Figure 3). Out of the 20 groups that trained explosive weight training with moderate loads, one group achieved very large ES (>1.2), three groups achieved a large ES (>0.80), five groups achieved a moderate ES (>0.5), while the remaining groups had a zero to low ES (0.01–0.5) (Figure 3). All the groups trained a form of squats. With the exception of two groups [61,69], all had a training frequency of a least twice a week. Most of the groups included a training period of six to seven weeks, whilst the remaining groups had a training period that included a longer training period of eight to ten weeks.

Out of the 20 groups, only two groups measured flying start at 5 to 20 m [63,67]. The percentage progression (Figure 4) was 0.93% [63] and 0.64% [67], respectively. The ES was very low to low (Figure 5). For flying start, the average percentage progression was 0.79% and the ES 0.17 (Table 5).

### 3.3. Explosive Weight Training with Light Loads (30–60% of 1-RM)

On average, explosive weight training with light loads demonstrated a percentage progression in the 10-m sprint of 1.70%, while the average ES (Table 5) was low (0.41). Percentage change (Figure 2) varied from −1.13% [77] to 4.62% [63]. The study with the highest percentage progression included a group of relatively untrained men who performed jumping squats at five repetitions x five sets, 30% of 1-RM. The ES of the groups varied from no ES to large ES (Figure 3). Fourteen of the 20 groups had a training frequency of at least two sessions a week. The majority included a training period of six to seven weeks, while the remaining groups were between eight and 12 weeks. With the exception of one group [75], all the groups had a training program that included a form of squat. Only one group achieved a large ES, nine groups had a moderate ES, while the remaining groups achieved zero to low ES after the training period ended (Figure 3).

Sixteen of the 20 groups measured a form of flying start between 5 to 20 m. Explosive weight training with light loads produced an average percentage progression of 1.07%, while the average ES (Table 5) was low at 0.27. Percentage change (Figure 3) varied from 0% [75,77,78,80] to 3.13% [79]. Four groups had a moderate ES, five had a low ES, and the remaining had zero to very low ES on flying start (Figure 5).

### 3.4. Hypertrophy Training (60–85% of 1-RM)

On average, hypertrophy training demonstrated the lowest percentage progression on the 10-m sprint with 0.47%, while the average ES (Table 5) was low at 0.21. The percentage decrease and progression (Figure 4) varied from −5.98% [86] to 2.54% [84]. The study with the most progression [84] utilized a periodical training program of squats and bench press that over time progressively increased from the lower to the upper portion of 60–90% of each participant’s 1-RM. All groups had a training frequency of at least two sessions a week, eight of the groups had a training period of four to six weeks, and the remaining had a period of eight to 12 weeks. All groups trained a form of squat apart from one group that trained hip thrusts [83]. ES varied from zero ES to large ES (Figure 3). One group had a large ES, two groups had a moderate ES, five had a low ES, and three had zero ES (Figure 3). Of the eleven groups that trained hypertrophy, only one measured a form of flying start [86]. Moir et al. [86] had the largest percentage progression (6.57%) and ES (1.13) of all the groups that measured flying start across all the different training methods (Table 5).

## 4. Discussion

The primary aim of this review was to determine which of the weight categories—maximal weight training, moderate explosive weight training, light explosive weight training, and hypertrophy training—produced the best effects on the first portion of the acceleration phase (10 m). The secondary aim was to determine which of the interventions also produced the best progression on flying start (10–40 m) within the selected studies that measured this over 10 m. The largest ES and percentage progression in the 10 m was found among the groups that performed maximal strength training. Maximal weight training also had the highest percentage progression [48,51] and ES [60]. Of the categories that included more than one measurement for flying start, explosive weight training with light loads had the highest average percentage progression, whilst maximal weight training had the highest ES. Viewed in isolation, Moir, et al. [86] had the largest ES and percentage progression; however, they also had the only experimental hypertrophy group to measure the flying start.

### 4.1. Maximal Weight Training for 10 m

Thirteen of the experimental groups that used maximal weight training achieved at least a large ES (>0.8), five groups had a low to moderate ES (0.2–0.8), and two groups had no progression. Across all the training methods, maximal weight training also had the most groups with a training period of at least eight weeks. This prolonged training period provides more time for greater neural and physical adaptations to take place [88]. A common trait among 12 of the 13 groups that achieved at least a large ES, was that they all had a background in ball sports (football, rugby or handball). Another possible explanation for the positive trends among athletes who carried out maximal weight training is that they were exposed to high physical demands in their technical and physical training that could have had a marked positive effect on speed [89,90,91,92,93,94]. The fact that ball sports contain a multitude of short-sequence sprints could have resulted in an optimization of intermuscular coordination tailored towards sprinting, compared to performing weight training in isolation [24]. The group with the large ES who not came from ball-based sports, consisted of athletes from mixed martial arts [54]. They had two sessions per week with maximal strength training, and a third session with explosive strength training with light loads. The combination of maximal strength training and explosive weight training with light loads could have led to improvements in the maximal development of force and RFD at different shortening speeds which led to a more optimal development of maximal power [95,96]. Additionally they trained exercises for jumping, sprinting and repeated sprints, which could give a more positive effect and optimization to intermuscular coordination [24]. This could explain the large ES for the group, even though they only had an intervention of four weeks.

Five out of the six groups that achieved huge ES (>2) were comprised of experimental groups with participants younger than 17 years of age. Younger people may experience a greater effect of maximal weight training because of experiencing a larger neuromuscular adaptation compared to older individuals who have more experience with training [97]. In addition, the younger groups could be made up of youths and adolescents who are developing late physically, which would mean they would have had a much greater improvement potential compared to youths and adolescents who developed early. This could have boosted the progression noted in the younger groups [98,99]. This was again demonstrated by Rodriguez-Rosell, et al. [78], who were able to demonstrate an inverse relationship between the rate of progression for speed and strength and the age of the individual (i.e., as an individual ages, his or her rate of progression for speed and strength decreases).

The three experimental groups that experienced low progression [58,59], the athletes were men who already trained with weights, meaning they likely had an elevated pre-study level of strength, thereby being more likely to experience a lower rate of progression compared to individuals who did not train strength prior to the study. A high level of maximal strength will set the limit for developing large power, and athletes who already have a high level of maximal strength should use other training methods in order to increase speed [33]. Furthermore, Weakley, et al. [59] only had an intervention of four weeks which give a lower progression due to less neural and physical adaptions [88]. One could argue that if the training dosage was too low, there likely would not have been enough stimuli to increase maximal strength beyond existing limits. A high training dosage is required for athletes with significant training experience to experience improvements in performance [35]. None of the athletes did any specific type of sprint training or took part in ball sports, which would have caused intermuscular coordination to be less optimal and less specific [24]. Weight training of the lower body often consists of vertical exercises, and, therefore, any potential improvement in RFD and/or maximal force development through weight training does not automatically translate to improved performance in a sprint where the aim is to develop horizontal force [6]. The same reasons can also be used to explain why the experimental groups of Jarvis, et al. [53] and Pedersen, et al. [55] did not experience any progression from maximal strength training. Another explanation for the lack of progress could be that there was too much stress due to overtraining/overload. The group in Jarvis, et al. [53] trained consistently five x five repetitions of 85% of 1-RM. An athlete’s ability to adapt to the stimulus of training is reduced when training resistance and intensity are too high over a prolonged period [100].

### 4.2. Explosive Weight Training with Moderate Loads for 10 m

One group showed very large ES, three groups a large ES, five groups a moderate ES, eight groups with low ES, and three groups with zero ES. The one group that experienced the very large ES [61], consisted of handball players, which is beneficial to achieve more stimulus for the lower body through the physical demands in the sports [93]. They trained one session per week with maximal weight training with half squats and leg press with 4 sets and 6 repetitions. Additionally, they also had another session with explosive weight training with light loads with the same exercises with 4 sets and 8 repetitions. The combination of light/heavy is more likely to improve the development of maximal force and RFD at different shortening speed, than only one intensity alone, and therefore result in a better maximal power development [95,96]. Also did they have a long intervention with 12 weeks, which provides more time for a better neural and physical adaption [88].

A common feature amongst seven of the nine experimental groups that noted the greatest progression was that they either came from ball sports and/or did specific sprint training additionally. The other two groups, that did not come from ball sports, or did specific sprint training, trained squats with full range of motion (ROM). Squats with a full ROM have been shown to provide better neuromuscular and functional adaptation compared to half squats [101].

Reasoning for low progression in different studies are several. Firstly, the short intervention period of six weeks [69], which provided a shorter time for neural and physical adaptions to occur [88]. Secondly, fatigue during training as observed in Griffiths, et al. [64]. In this study the group trained RPE10 on every set, which is until full exhaustion. Explosive weight training with moderate loads should be performed with low repetitions and stop before RPE7 to manage the fatigue, while getting physiological and performance adaptions [95,96,102]. Thirdly, interference from aerobic training in the same session [70]. Previous research have suggested that concurrent training with aerobic training and strength training could impair the ballistic and strength adaptions [103]. Another reason is the fact that they could have trained with higher intensity to get further strength adaptions and could possibly counter the negative effects of the aerobic training. This was shown by the other group in the same study who had a moderate progression with a higher intensity [70]. The trend of concurrent training with strength and aerobic training is again shown by Sousa, et al. [71], where the three groups differenced from low, moderate and large progression. That study showed an inverse relationship between aerobic intensity and progression on 10-m sprint, where higher aerobic intensity gave less progression on acceleration, even though the three groups trained on the same intensity in squats with 70–85% of 1-RM. Fourthly, the lower rate of progression could also be due to groups consisting of well-trained, physically fit participants, who therefore may have benefited from a higher training volume [62,66]. A higher total training dosage is required for athletes who have a relatively high level of previous training experience [35]. Lastly, the lack of progression in the study by Cormie, et al. [63] is likely attributed to the participants consisting of relatively untrained men and the level of resistance and training volume over time being too great, thereby causing them to become overtrained and impairing their ability to adapt physically and neurologically to exercise. Furthermore, untrained individuals have poorer recovery times. When considering that they had a high training dosage that included a frequency of three sessions a week with a resistance level of 75%, 80%, and 90% of 1-RM, this could have placed too much strain on the musculoskeletal and neurological systems to the point where there was no measurable progression in sprinting performance [100].

### 4.3. Explosive Weight Training with Light Loads for 10-m Sprint

Explosive weight training with light loads provided large ES in one experimental group and moderate ES in nine of the experimental groups. The group that had a good ES consisted of football players who, in addition to the weight training, were also training sprinting and jumping exercises, which would have contributed to providing a more specific adaption relevant to sprinting performance [24]. The same can be said of the five other groups where the participants also had a ball-sports background. The explanation as to why the ES in general among the groups is moderate could be associated with how explosive weight training with light loads primarily trains an athlete to improve RFD at a lighter level of resistance, so that speed during movement throughout the repetitions is more specific and like the speed utilized in a sprint [25]. This type of resistance, therefore, primarily provides neural adaptation through improved firing frequency, neural activation and intermuscular coordination [25]. The lack of any significant metabolic stress and mechanical resistance means that muscle growth would be minimal. Therefore, training would not affect the development of maximal force, especially if the training intensity was under 50% of 1-RM [35]. In addition, strength gains would also be minimal, as minimal strength gains would be expected from training at a dosage of 50–70% of 1-RM, while increasing training resistance above that level would have provided an increase in strength [35]. This could also explain why the most groups experienced a low ES on 10-m performance after explosive strength training with light loads had ended. Pareja-Blanco, et al. [77] found very low ES on 10-m performance after the intervention, which consisted of physically active men who trained squats with intensity between 40–55% of 1-RM.

Three more groups who did not register any ES on 10-m performance after explosive weight training with light loads [74,76,81]. The groups [74,76], Yáñez-García, et al. [81] consisted of male football players and athletes from basketball between the ages of 16 and 19 years old. A possible explanation for the lack of progression is that they had a high pre-existing maximal power production, which they could have received from previous ball-based training, as football and basketball is a sport that demands the development of and application of a high level of force from the lower body fast across a variety of movements [89,90,92,94] Thus, football and basketball players already would have had a high level of previous training experience with the type of force development that is carried out in explosive weight training with light loads, so they would have required a higher training dose to register any progression [35]. The limitations of the two groups are unlikely related to how good they were at developing maximal power quickly and often, but rather to a lack of strength required to develop high maximal power. The ability to generate force rapidly is of little benefit when the development of maximal force is low [102]. Therefore, the level of maximal strength will dictate how much maximal power a person can develop [25]. It would be reasonable to conclude that a higher training dose at 80–89% of 1-RM for young athletes is required in order to develop maximal strength [104]. Training that increases maximal strength increases the capacity to develop power [105].

### 4.4. Hypertrophy Training for 10 m

Among the eleven experimental groups that performed weight training without the intention of maximal speed during the concentric phase, only one group registered a large ES. That group was, in return, one of two groups that adopted the longest training period of 12 weeks with two sessions a week [84], where they utilized linear periodization of training that ranged from 60% to 90% of 1-RM over 12 weeks. Periodical training that follows a progressive model can significantly increase strength and lead to hypertrophy over a 12-week training period [36,106]. In addition, the participants consisted of 16–17-year-old rugby players [84]. Younger people can experience a larger effect because of a greater neuromuscular adaptation compared to adults who have more training experience [97]. An increased maximal strength through hypertrophy and neural adaptations provide a higher capacity and potential to develop maximal power [105].

Of the remaining seven groups that recorded progressions in ES, two groups had a moderate ES, while the remining had low ESs, even though four of the experimental groups [84,85,87] contained athletes from ball sports, which provide an additional stimulus for specific adaptation [89,90,91,92,93,94]. The low progression that arises because of hypertrophy training could be a result of training being carried out without intending to achieve maximal speed. As mentioned previously, repetitions carried out with maximal speed during the concentric phase will provide better improvement in maximal strength and power compared to repetitions without the intention for maximal speed [107,108]. The intention of carrying out repetitions explosively is meant to provide better RFD through neural adaptation at a given resistance level [25]. The theory explains how adaptations in force become specific to the speed of movement that is carried out during training [95]. If the participants end up performing weight training repetitions without the intention for maximal speed, there will be zero to minimal development of RFD in sprinting. With regard to the force–velocity curve by Hill [34], any potential improvement in performance over 10 m will first and foremost be related to increasing maximal strength through hypertrophy and neural adaptations associated with an increase in strength.

Three experimental groups had no ES after the training period [82,83,86]. Two of the groups consisted of active men under 20 years old, and one group had a background in rowing, whilst the remaining group consisted of women who played basketball. The group in the study of Moir, et al. [86] consisted of active men who took part in a variety of ball sports. However, during the study they were asked to refrain from any other activities that included sprinting. During an eight-week period without training in ball sports that include sprinting, physical deconditioning is likely to take place where there is a loss of physical capacity because of a reduction or complete absence of training stimulus [109]. Physiologically speaking, there is a loss of intermuscular coordination required for the specific movements that are carried out during a sprint [24]. This process of deconditioning over time could explain why the group in study of Moir, et al. [86], which previously had a certain level of capacity within sprinting prior to joining the study, measured a regression during the study. The regression may not have been because of poor effects from weight training, but rather that they were no longer being stimulated in an activity they regularly took part in prior to commencing the study. In the study by Moir, et al. [86], the group also had a longer contact time during the first three steps of sprinting, which likely means a decline in step frequency occurred, which is not beneficial to the improvement of acceleration [42,43].

The lack of progress noted by Contreras, et al. [83] could be a result of insufficient neural adaptation from training that did not have the intention of maximal speed [25]. The group did not train sprinting before or during the intervention, so no coordination of the movements required for high performance in sprinting took place. In other words, the potential to accelerate faster could have occurred, but with no intermuscular coordination through specific sprint training, there may be a lack of progress [24]. As previously stated, an increase in vertical strength does not always provide an increase in horizontal force development, especially if specific movement patterns relevant to sprinting are not stimulated during training [6].

The group that consisted of women were active basketball players [82], did squats at 65% of 1-RM with 4 sets and 8 repetitions. The intervention lasted 4 weeks with a training frequency at 2 times per week. The lack of progressive overload or periodization can be one reason to explaining why the participants did not have any progression and transfer to 10-m sprint performance. Even though physiological adaptations can occur in a resistance program with no variation or progression over a relatively short time period, is it necessary to have a systematic increase in the demands placed upon the body for further adaptions [110]. The repetitions were also executed with 4 s in the eccentric phase and two seconds in the concentric phase of the lift. Which means the intention to lift without maximal speed could result of insufficient neural adaption, as mentioned before [25]. Lastly, the intervention of 4 weeks could been too short for neural and physical adaptions to take place [88].

### 4.5. Weight Training for a Flying Start

During a flying start, the ES of maximal weight training ranged from zero to moderate, with the highest average ES among the categories. The lower ES on flying start could be attributed to the fact that acceleration and near maximal speed are two specific and individual factors [111]. A lower level of specific adaptations could also have occurred with fewer sprints at near maximal speed, compared to sprints that focused on acceleration in the studies that had participants who trained in ball sports and sprint training [24]. During the first 10 m of an acceleration, the maximal force development in relation to body weight will play a significant role, as shortening speed is slower and force development is higher during the initial portion of a sprint [34,111]. Therefore, maximal weight training will have the greatest influence.

When sprinting at near maximal or maximal speed, the shortening speed is high, while force is low [34,111]. The improvement that maximal weight training could have provided is related to force becoming higher at the same shortening speed for the groups that experienced a progression. This could also have been the case in the two groups that performed explosive weight training with moderate loads, where the ES was very low to low [63,67]. In this situation, potential improvements to RFD were specifically adapted to the resistance or movement they were practicing [95].

In the groups that performed explosive weight training with light loads, there were far more studies that measured flying starts, where four of sixteen experimental groups had moderate ES after the intervention period. High-speed strength training below 60% of 1 RM is almost just as effective as the maximal weight training on flying start, and more effective compared to explosive weight training with moderate loads. This is explained by the fact that the participants are training at a shortening speed that is like the shortening speed they have during their maximal speed in a sprint [34,111]. It is likely the RFD that is improving at similar shortening speeds that they have during near maximal or maximal speeds during a sprint through neural adaptations of firing frequency and intermuscular coordination [25]. Another four groups did have low ES, and the remaining groups varied from very low to no ES. A possibly explanation to the low progress in the groups could have been the limit of the maximal power production. With higher velocity, increases the demands for power output [22]. The maximal strength limits the production of maximal power. This explains, that the participants could possibly have gained more on flying start with focusing on maximal weight training to increase the maximal strength, and therefore, increase the production of maximal power as shown by the highest average ES increase [25].

Moir, et al. [86] was the only one to measure flying start in the hypertrophy group. It was also the only group in any category that noted a large ES on flying start. The explanation for the progression is likely due to the inclusion of power cleans, an exercise found in weightlifting that requires participants to accelerate quickly using their whole body to counteract the resistance they are lifting. The force to counteract the resistance must be high and leads to a natural speed development through the entire ROM [25]. Power cleans are therefore an exercise that demand a high level of movement speed, and it has been shown that the maximal power is highest at 80% of 1-RM in power cleans [112].

Maximal running speed at the same time is also influenced by reactive strength [111]. Reactive strength is a muscle’s ability to quickly produce the force that arises during the eccentric phase to quickly utilize it in the concentric phase during a stretch–shortening cycle, for example, during contact time in the steps of a sprint [111]. Therefore, explosive plyometric training should also be included in a training program intended to improve maximal running speed [111].

### 4.6. Practical Consequences

After calculating both ES and progression percentage, this review has demonstrated that maximal weight training is the best method to improve acceleration. This is likely associated with the fact that most of the participants did not have a particularly high level of strength before joining their respective studies, so the large increase in strength after the intervention period led to an improvement in absolute and relative force production, thereby improving acceleration. The maximal power that is considered beneficial for both acceleration and maximal speed is dictated by the ability to generate maximal force [25]. Acceleration demands a higher level of force production initially, and the closer the athlete gets to maximal speed, the lower the level of force required to reach maximal speed. This is explained by the Hill curve [34], where force is greater at a lower contraction speed and becomes lower when the contraction speed becomes faster. This also explains why explosive weight training with light loads showed a similar effect compared to maximal weight training on flying start, as the athletes who are near maximal speed require much faster development of force.

Weight training that targets speed assumes a higher degree of individualization with regards to physical fitness ability and age. Better trained athletes with a high relative strength level will likely have a greater need to train at different intensities below 85% of 1-RM from very light intensity (0–40% of 1-RM), light intensity (40–60% of 1-RM), and medium intensity (60–85% of 1-RM) with maximal speed to experience a significant progression in sprinting performance [113,114,115,116]. Research has shown that individuals with a high level of relative strength do not experience any progression in their sprinting ability after eight weeks of intervention with a heavy weight training program compared to individuals with a low relative strength who did see progression in sprinting from the same intervention [102]. Nevertheless, a weight training program must consider how much weight training the athlete can handle to achieve the right dose–response relationship to get the most out of his or her weight training sessions. It is also important to consider that athletes with a higher relative strength in the lower body recover faster and can handle a higher training load than athletes with a lower relative strength [88,116]. The same principle can be said for athletic populations that can handle a higher training volume compared to non-athletic populations [35]. Older youth athletes also need to train more heavily than younger youth athletes, independent of their previous experience within resistance training, to achieve similar gains in sprinting, jumping, and strength [78]. Furthermore, full ROM in the exercises is important because it can lead to better neuromuscular and functional adaption than exercises completed with more limited ROM [101]. The range of exercises can also affect progress. Some exercises like power cleans require naturally rapid development of force, which makes an exercise like power cleans more specific than squat in relation to acceleration and close to the maximum speed in a sprint [112].

For athletes who take part in sports where short sprints are important, weight training can be recommended, particularly maximal weight training, to improve acceleration. For athletes who already possess a high level of strength, the focus should be on maintaining strength while seeking other training programs such as moderate and especially explosive training with light loads to specifically improve RFD at a higher shortening speed [25]. At the same time, considerations must be made to avoid training too heavily or over too long a period to avoid overtraining and injury [100]. In the long run, weight training should be performed two to three times a week using a periodized program that includes maximal, moderate, and light explosive training for the maximal development of force and RFD at different shortening speeds to allow for optimal development of maximal power [95,96]. Weight training will then improve the potential to achieve higher acceleration and maximal running speed through improved maximal strength in relation to body weight and improved RFD. To achieve the best effects, it is necessary for weight training to include specific adaptations related to sprinting [24]. It is also worth noting that strength training should, as a principle, be a secondary intervention, as it is the specific movement stimulus an athlete achieves in sprinting that first and foremost improves sprinting performance [6].

It has to be noticed that the present revew has some limitiations. Firstly, the different weight training interventions were categorised after different relative loads and did not investigate the effect of the different and number of weight exercises used (squats, hip trusts, etc.). Secondly, the total work load (weeks of training, number of repetitions and sets, etc.) of each intervention was not discussed in the review, which can have a large effect upon the findings related to the sprint performance. Thirdly, different samples (men and women) from different sports with different performance were investigated. All these extra parameters make it very difficult to calculate the total workload of each intervention and the effect of this on sprint in performance in different samples (soccer, rugby, rowing, basketball, etc.). Therefore, the reader must be aware of these shortcomings when using the findings of the present review.

## 5. Conclusions

Both acceleration and near maximal speed are important attributes in many sports. As speed is affected by muscle strength, it has been important to determine what effect the various weight training interventions have on acceleration and near maximal speed. At 10 m, all weight training interventions demonstrated improvements, and maximal weight training demonstrated the highest results with a large ES, while other programs varied from very small to moderate ES. Weight training showed little progression with a significantly lower effect on flying start across all training methods, except for one group that trained power cleans (hypertrophy) where progress was large. To improve acceleration, this review has demonstrated maximal weight training as the preferred training. For athletes with a pre-existing high level of strength, it could be more appropriate to use explosive training with light loads or a combination of the two. To a lesser extent, acceleration from a flying start could be improved using both training programs as well.

## Figures and Tables

**Figure 1 sports-10-00148-f001:**
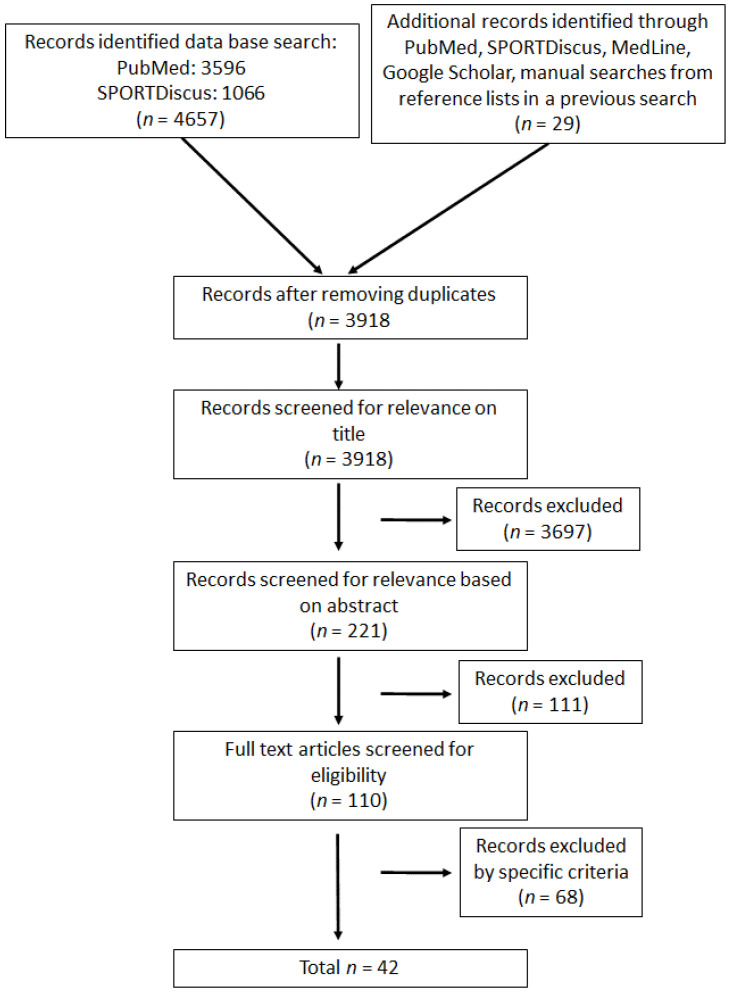
A schematic representation of the searching process to find eligible studies for this review. A PRISMA flowchart was used to the illustrate inclusion and exclusion criteria used in this review.

**Figure 2 sports-10-00148-f002:**
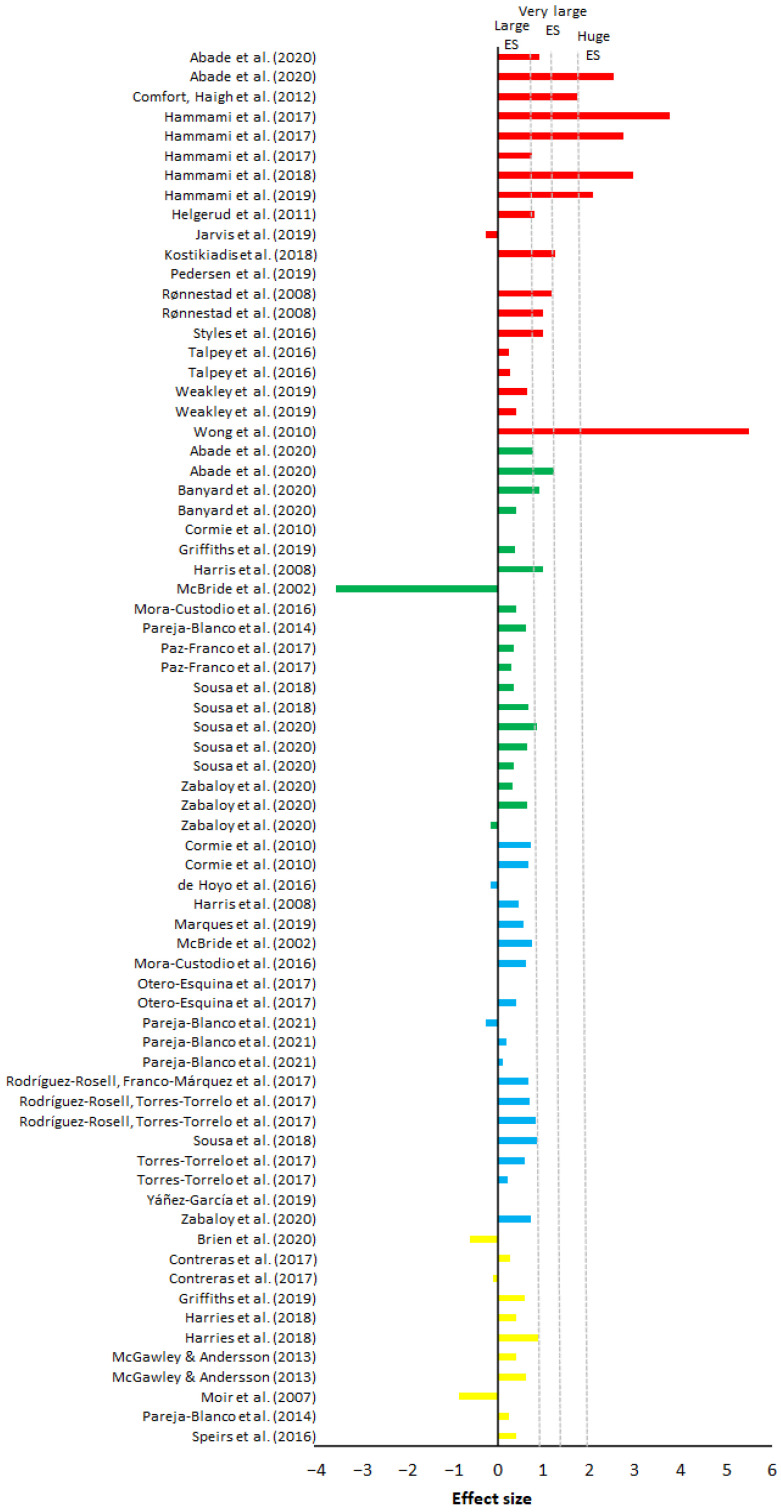
Percentage of change (%) of each maximal weight training (red), explosive weight training with moderate loads (green), explosive weight training with light loads (blue), and hypertrophy training (yellow) studies on 10-m sprint times.

**Figure 3 sports-10-00148-f003:**
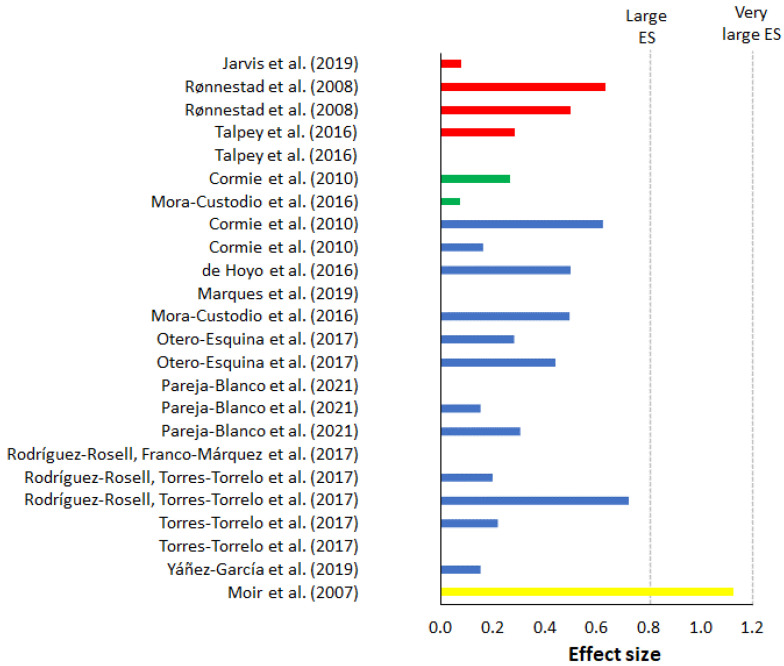
ESs of maximal weight training (red), explosive weight training with moderate loads (green), explosive weight training with light loads (blue), and hypertrophy training (yellow) studies on 10-m sprint times.

**Figure 4 sports-10-00148-f004:**
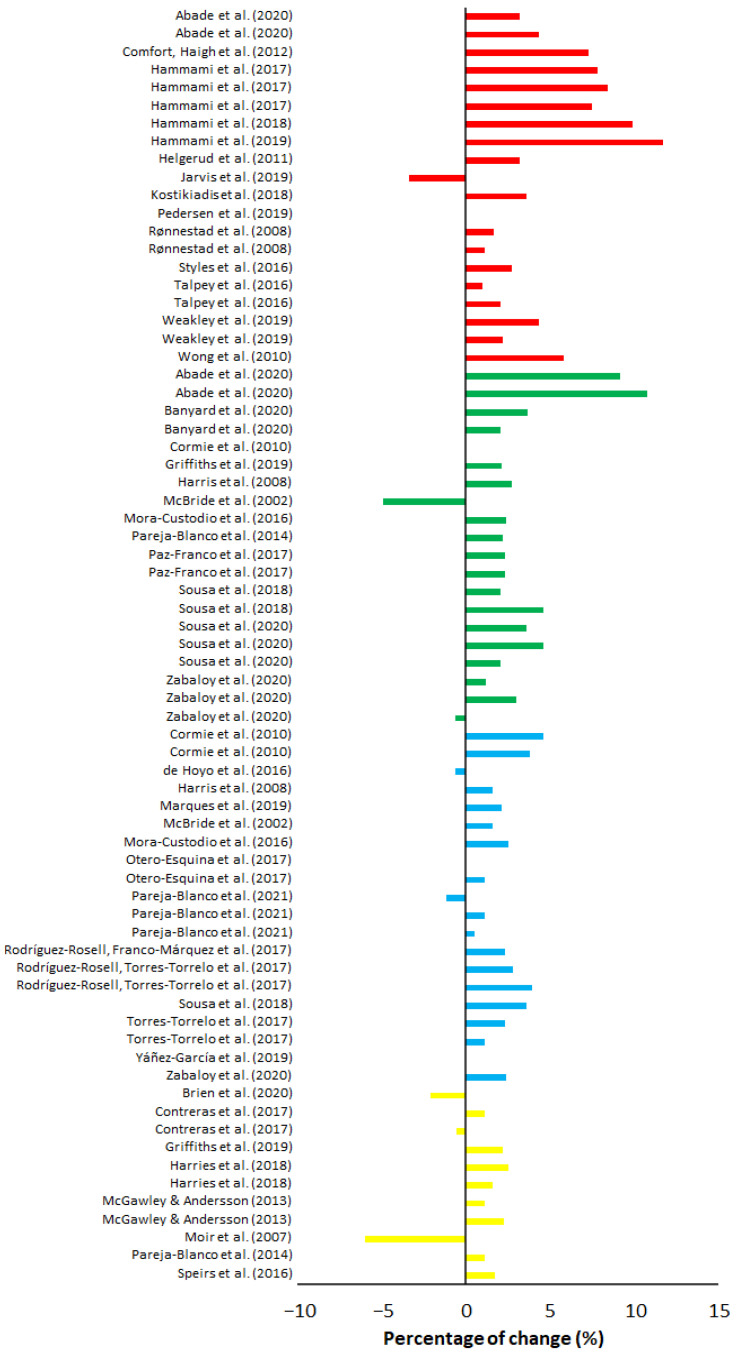
Percentage of change (%) of each maximal weight training (red), explosive weight training with moderate loads (green), explosive weight training with light loads (blue), and hypertrophy training (yellow) studies on flying 10-m sprint times.

**Figure 5 sports-10-00148-f005:**
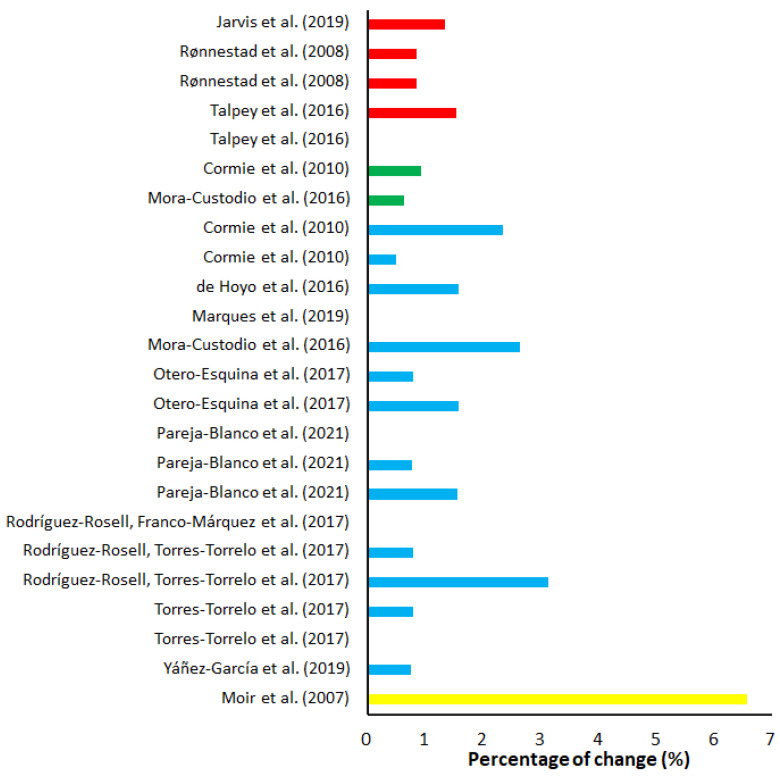
ESs of maximal weight training (red), explosive weight training with moderate loads (green), explosive weight training with light loads (blue), and hypertrophy training (yellow) studies on flying 10-m sprint times.

**Table 1 sports-10-00148-t001:** Maximal weight training interventions (85–100% of 1-RM).

Reference	Participant/Age (Years)	Level	Weeks/Week Sessions	Intervention (Sets/Reps) (% av 1-RM)	Other Training/Exercises	Percentage Change %	Effect Size (ES)
Abade, et al. [46]	8 men, 16.56 + 0.56	Football	20/1	Half squats, 3 sets/4–10 RM	General strength training in same session, football	10 m: 3.23	10 m: 0.92
Abade, et al. [46]	8 men, 16.56 + 0.56	Football	20/1	Hip thrust, 3 sets/4–10 RM	General strength training in same session, football	10 m: 4.35	10 m: 2.53
Comfort, et al. [47]	19 men, age not stated	Rugby elite	8/2	Squats, midthigh clean pull, Romanian deadlifts, Nordic curls, hang cleans, squat jumps, 4 sets/6 reps 85–90%	Agility, sprint, rugby	10 m: 7.3	10 m: 1.65
Hammami, et al. [48]	17 men, 16.0 ± 0.5	Football	8/2	Half squats, 3 sets/8 reps (70%), 5 sets/4 reps (80%), 4 sets/3 reps (85%), 3 sets/3 reps (90%)	Sprint and jumping training after every set of strength exercise, football	10 m: 11.68	10 m: 2.08
Hammami, et al. [49]	16 men, 16.2 ± 0.6	Football	8/2	Half squats, 3 sets/8 reps (70%), 5 sets/4 reps (80%), 4 sets/6 reps (85%), 3 sets/3 reps (90%)	Football	10 m: 8.42	10 m: 2.74
Hammami, et al. [49]	16 men, 16.0 ± 0.5	Football	8/2	Half squats, 3 sets/8 reps (70%), 5 sets/4 reps (80%), 4 sets/6 reps (85%), 3 sets/3 reps (90%)	Sprint and jumping training after every set of strength exercise, football	10 m: 7.49	10 m: 0.75
Hammami, et al. [50]	19 men, 16.2 ± 0.6	Football	8/2	Half squats, 3 sets/8 reps (70%), 5 sets/4 reps (80%), 4 sets/3 reps (85%), 3 sets/3 reps (90%)	Football	10 m: 9.90	10 m: 2.97
Hammami, et al. [51]	14 women, 16.6 ± 0.3	Handball	10/2	Half squats, leg press, calf extension, 4 sets/6 reps (85%) and isometric half squats 4 sets/8 reps (75%)	Sprint and jumping training after every set of strength exercise, handball	10 m: 7.85	10 m: 3.75
Helgerud, et al. [52]	21 men, 25.0 + 2.9	Football, elite	8/2	Half squats, 4 sets/4 reps (85–90%)	Cardiovascular conditioning, football	10 m: 3.21	10 m: 0.8
Jarvis, et al. [53]	11 women and men, 27.36 ± 3.17	Collegiate athletes	8/2	Hip thrust, 5 sets/5 reps (85%)		10 m: −3.33 10–20 m: 1.33	10 m: −0.24 10–20 m: 0.08
Kostikiadis, et al. [54]	10 men, 28.9 + 4.2	Mixed martial arts	4/2	Squats, deadlifts, 3–5 sets/2–8 reps (80–95%)	jump squats with 4 sets/8 reps with 30% of 1-RM, rowing, upper body strength exercises, jumping, speed drills, repeated sprints	10 m: 3.59	10 m: 1.27
Pedersen, et al. [55]	19 women, 18.0 + 3.0	Football	5/2	Squats 3–4 sets/4–6 reps (>85%), Nordic hamstring 3 × 6	Football	10 m: 0	10 m: 0
Ronnestad, et al. [56]	6 men, 22.0 ± 2.5	Football	8/2	Half squats, hip flexion 3–5 sets of 4–6 RM	Football	10 m: 1.68 30–40 m: 0.85	10 m: 1.18 30–40 m: 0.63
Ronnestad, et al. [56]	8 men, 23.0 ± 2.0	Football	8/2	Half squats, hip flexion 3–5 sets of 4–6 RM	Jumping training, football	10 m: 1.14 30–40 m: 0.85	10 m: 1 30–40 m: 0.5
Styles, et al. [57]	17 men, 18.3 ± 1.2	Football	6/2	Squats, Romanian deadlifts, 4 sets/5 reps (85–90%) and 3 sets/3 reps (85–90%) and Nordic lowers (body weight)	Football	10 m: 2.73	10 m: 1
Talpey, et al. [58]	9 men, 21.44 ± 3.54	Weights training	9/2, pause week 4 and 5	Squats, 3–4 sets/3–6 reps of 3–8 RM, jumping squats after squats		10 m:1.02 15–20 m: 1.54	10 m: 0.24 15–20 m: 0.28
Talpey, et al. [58]	11 men, 20.91 ± 3.59	Weight training	9/2, pause week 4 and 5	Squats, 3–4 sets/3–6 reps of 3–8 RM, jumping squats before squats		10 m: 2.04 15–20 m: 0	10 m: 0.28 15–20 m: 0
Weakley, et al. [59]	16 men, 21.0 + 1.0	Rugby	4/2	Squats, hex bar deadlift, 3 sets/2–5 reps (85–93%), Nordic curl, glute bridges	Rugby, Squat jumps with 3 sets/3 reps (20%) performed same session, other exercises for core and upper body, sprinting, jumping	10 m: 4.35	10 m: 0.64
Weakley, et al. [59]	12 men, 21.0 + 2.0	Rugby	4/2	Squats, hex bar deadlift 3 × 2–5 85–93% 1 RM, Nordic curl, glute bridges	Rugby, squat jumps with 3 sets/3 reps (20%) performed same session other exercises for core and upper body, sprinting, jumping	10 m: 2.21	10 m: 0.4
Wong, et al. [60]	20 men, 24.6 ± 1.5	Football, elite	8/2	High pull, half, and jump squats, bench press, chin-up 4 sets/6 RM	Cardiovascular conditioning, football	10 m: 5.82	10 m: 5.5

**Table 2 sports-10-00148-t002:** Explosive weight training with moderate load interventions (60–85% of 1-RM).

Reference	Participant/Age (Years)	Level	Weeks/Week Sessions	Intervention (Sets/Reps) (% av 1-RM)	Other Training/Exercises	Percentage Change %	Effect Size (ES)
Abade, et al. [61]	10 men, 24.7 + 3.8	Handball	12/2	Half squats, leg press, bench press, 2 sets/6 reps (80%)	Same exercises performed same session with 2 × 8 (30%), Handball	10 m: 9.14	10 m: 0.78
Abade, et al. [61]	10 men, 24.7 + 3.8	Handball	12/1	Half squats, leg press, bench press, 4 sets/6 reps (80%)	Same exercises on a different day with 4 × 8 (30%), Handball	10 m: 10.81	10 m: 1.24
Banyard, et al. [62]	12 men, 25.5 ± 5.0	Train with weights	6/3	Squats, 5 sets/5 reps (20–90%) (avg: 69.2%)		10 m: 3.68	10 m: 0.93
Banyard, et al. [62]	12 men, 26.2 ± 5.1	Train with weights	6/3	Squats, 5 sets/5 reps (59–85%) (avg: 70.9%)		10 m: 2.11	10 m: 0.4
Cormie, et al. [63]	8 men, 23.9 ± 4.8	Relatively untrained	10/3	Jumping squats, SE 1: 3 sets/3 reps (90%) SE 2: 3 sets/6 reps (75%) SE 3: 3 sets/4 reps (80%)		10 m: 0.5 20 m: 0.93	10 m: 0.5 20 m: 0.27
Griffiths, et al. [64]	15 men, 23.0 + 1.0	Football	6/2	Squats, knee extension, knee flexion, hip extension, hip flexion, heel raises 3 sets/performed to RPE10 (80%)	Strength exercises upper body, Football	10 m: 2.16	10 m: 0.38
Harris, et al. [65]	9 men, 21.8 ± 4.0	Rugby, elite	7/2	Jumping squats, 5 sets/5 reps (80%)	Sprint training, rugby	10 m: 2.73	10 m: 1
McBride, et al. [66]	10 men, 21.6 ± 0.8	Train with weights	8/2	Jumping squats, 4 sets (80%)		10 m: −4.89	10 m: −3.53
Mora-Custodio, et al. [67]	9 women, 22.4 ± 1.9	Physically active	12/2	Squats, 3 sets/4–6 reps (65–80%)		10 m: 2.42 10–20 m: 0.64	10 m: 0.42 10–20 m: 0.07
Pareja-Blanco, et al. [68]	10 men, 23.3 ± 3.2	Train with weights	6/3	Squats, 3–4 sets/2–8 reps (60–80%)		10 m: 2.78	10 m: 0.77
Paz-Franco, et al. [69]	12 men, 23.7 + 6.1	Futsal, elite	6/2	Half squats, leg press, hamstring curl, 75% 3 sets/8 reps	Futsal	10 m:2.36	10 m: 0.36
Paz-Franco, et al. [69]	12 men, 23.6 + 5.7	Futsal, elite	6/1	Half squats, leg press, hamstring curl, 75% 3 sets/8 reps	Futsal	10 m: 2.36	10 m: 0.3
Sousa, et al. [70]	9 men, 20.6 + 1.9	Physically active	8/2	Squats, 3 sets/5–8 reps (70–85%)	Sprinting, jumping, aerobic training	10 m: 3.21	10 m: 0.67
Sousa, et al. [70]	9 men, 20.6 + 1.6	Physically active	8/2	Squats, 3 sets/6–8 reps (55–70%)	Sprinting, jumping, aerobic training	10 m: 1.09	10 m: 0.31
Sousa, et al. [71]	10 men, 21.2 + 1.5	Physically active	8/2	Squats, 3 sets/5–8 reps (70–85%)	Sprinting, jumping, aerobic training	10 m: 3.59	10 m: 0.88
Sousa, et al. [71]	10 men, 21.0 + 2.0	Physically active	8/2	Squats, 3 sets/5–8 reps (70–85%)	Sprinting, jumping, aerobic training	10 m: 4.62	10 m: 0.64
Sousa, et al. [71]	10 men, 21.1 + 2.2	Physically active	8/2	Squats, 3 sets/5–8 reps (70–85%)	Sprinting, jumping, aerobic training	10 m: 2.04	10 m: 0.36
Zabaloy, et al. [72]	11 men, 23.73 ± 3.32	Rugby, elite	7/2	Squats, 3–4 sets/3–6 reps (75–85%) and jumping squats	Rugby, sprinting, jumping	10 m: 1.18	10 m: 0.33
Zabaloy, et al. [72]	9 men, 22.00 ± 3.77	Rugby, elite	7/2	Squats, 3–4 sets/3–6 reps (60–75%) and jumping squats	Rugby, Sprinting, jumping	10 m: 2.99	10 m: 0.65
Zabaloy, et al. [72]	8 men, 21.43 ± 2.51	Rugby, elite	7/2	Squats, 3–4 sets/3–6 reps (60–75%) and jumping squats	Rugby, Sprinting, jumping	10 m: −0.60	10 m: −0.13

**Table 3 sports-10-00148-t003:** Explosive weight training with light load interventions (30–60% of 1-RM).

Reference	Participant/Age (Years)	Level	Weeks/Week Sessions	Intervention (Sets/Reps) (% av 1-RM)	Other Training/Exercises	Percentage Change %	Effect Size (ES)
Cormie, et al. [63]	8 men, 23.9 ± 4.8	Relatively untrained	10/1	Jumping squats, 5 sets/5 reps (30%)	Two other sessions a week of jumping squats 0% of 1-RM	10 m: 4.62 5–20 m: 2.34	10 m: 0.73 5–20 m: 0.63
Cormie, et al. [73]	8 men, no age stated	Trains with weights	10/1	Jumping squats, 5 sets/5 reps (30%)	Two other sessions a week of jumping squats 0% of 1-RM	10 m: 3.85 5–20 m: 0.51	10 m: 0.69 5–20: 0.16
de Hoyo, et al. [74]	11 men, 18 ± 1	Football	8/2	Squats, 3–6 sets/6 reps (40–60%) and leg curl	Football	10 m: −0.60 10–20 m: 1.57	10 m: −0.15 10–20 m: 0.5
Harris, et al. [65]	9 men, 21.8 ± 4.0	Rugby, elite	7/2	Jumping squats, 6 sets/10–12 reps optimal resistance (≈26.3%)	Sprint training, rugby	10 m: 1.61	10 m: 0.46
Marques, et al. [75]	11 men, 18.1 + 0.8	Futsal	6/2	Leg press 2–3 sets/5–6 repetitions (45–65%)	Futsal, sprint, cod, jump training	10 m: 2.15 10–20 m: 0	10 m:0.57 10–20 m: 0
McBride, et al. [66]	9 men, 24.2 ± 1.8	Trains with weights	8/2	Jumping squats, 5 sets (30%)		10 m: 1.57	10 m: 0.75
Mora-Custodio, et al. [67]	10 women, 22.4 ± 1.9	Physically active	12/2	Squats, 3 sets/4–6 reps (40–60%)		10 m: 2.54 10–20 m: 2.63	10 m: 0.62 10–20 m: 0.5
Otero-Esquina, et al. [76]	12 men, 17.0 ± 1.0	Football	7/1	Squats, 3 sets/4–6 reps (40–50%) and leg curl	Sprint with resistance, jumping, football	10 m: 0 10–20 m: 0.79	10 m: 0 10–20 m: 0.28
Otero-Esquina, et al. [76]	12 men, 17.0 ± 1.0	Football	7/2	Squats, 3 sets/4–6 reps (40–50%) and leg curl	Sprint with resistance, jumping, football	10 m: 1.17 10–20 m: 1.59	10 m: 0.4 10–20 m: 0.44
Pareja-Blanco, et al. [77]	15 men, 21.8 + 1.9	Sports science students	8/1	Squats, 3–4 sets/6–8 reps (40–55%)		10 m: −1.13 10–20 m: 0	10 m: −0.27 10–20 m: 0
Pareja-Blanco, et al. [77]	18 male, 22.8 + 3.0	Sports science students	8/1	Squats, 3–4 sets/6–8 reps (40–55%)	Sprint with heavy sled towing	10 m: 1.11 10–20 m: 0.77	10 m: 0.19 10–20 m: 0.15
Pareja-Blanco, et al. [77]	18 male, 22.6 + 2.8	Sports science students	8/1	Squats, 3–4 sets/6–8 reps (40–55%)	Sprint with light sled towing	10 m:0.56 10–20 m:1.55	10 m:0.11 10–20 m: 0.31
Rodriguez-Rosell, et al. [78]	14 men, 16.4 ± 0.5	Football	6/2	Squats, 2–3 sets/4–8 reps (45–60%)	Sprinting, jumping, turning/cutting speed, football	10 m: 2.33 10–20 m: 0	10 m: 0.67 10–20 m: 0
Rodriguez-Rosell, et al. [79]	10 men, 24.5 ± 3.4	Football	6/2	Squats, 2–3 sets/4–6 reps (45–60%)	Core exercises, football	10 m: 2.82 10–20 m: 0.78	10 m: 0.71 10–20 m: 0.2
Rodriguez-Rosell, et al. [79]	10 men, 24.5 ± 3.4	Football	6/2	Squats, 2–3 sets/4–6 reps (45–60%)	Sprinting, jumping, turning/cutting speed, football	10 m: 3.93 10–20 m: 3.13	10 m: 0.82 10–20 m: 0.72
Sousa, et al. [70]	8 men, 20.6 + 0.9	Physically active	8/2	Squats, 3 sets/6–8 reps 40–55%	Sprinting, jumping, aerobic training	10 m: 1.60	10 m: 0.3
Torres-Torrelo, et al. [80]	12 men, 23.8 ± 2.4	Futsal	6/2	Squats, 2–3 sets/4–6 reps (45–58%)	Futsal	10 m: 2.33 10–20 m: 0.79	10 m: 0.6 10–20 m: 0.22
Torres-Torrelo, et al. [80]	12 men, 22.9 ± 5.1	Futsal	6/2	Squats, 2–3 sets/4–6 reps (45–58%)	Turning/cutting speed, futsal	10 m: 1.16 10–20 m: 0	10 m: 0.23 10–20 m: 0
Yáñez-García, et al. [81]	11 subjects, 16.5 + 0.5	Basketball	6/2	Squats, 2–3 sets/4–8 reps (45–60%)	Basketball, sprinting, jumping, change of direction	10 m: 0 10–20 m: 0.76	10 m:0 10–20 m: 0.15
Zabaloy, et al. [72]	6 men, 21.5 ± 3.53	Rugby, elite	7/2	Squats 3–6 sets/3–4 reps (40–60%) and Jumping squats	Rugby, sprinting, jumping	10 m: 2.41	10 m: 0.72

**Table 4 sports-10-00148-t004:** Hypertrophy training interventions (60–85% of 1-RM).

Reference	Participant/Age (Years)	Level	Weeks/Week Sessions	Intervention (Sets/Reps) (% av 1-RM)	Other Training/Exercises	Percentage Change %	Effect Size (ES)
Brien et al. [82]	9 women, 24.18 + 6.56	Basketball	4/2	Squats 4 sets/8 reps (65%)	Basketball	10 m: −2.06	10 m: −0.6
Contreras, et al. [83]	14 men, 15.49 ± 1.16	Rowing	6/2	Hip thrust 4 sets/6–12 RM		10 m: 1.14	10 m: 0.27
Contreras, et al. [83]	14 men, 15.48 ± 0.74	Rowing	6/2	Squats 4 sets/6–12 RM		10 m: −0.56	10 m: −0.1
Griffiths et al. [64]	15 men, 21.0 + 1.0	Football	6/2	Squats, knee extension, knee flexion, hip extension, hip flexion, heel raises 3 sets/performed to RPE10 (80%)	Strength exercises upper body, Football	10 m: 2.19	10 m: 0.59
Harries, et al. [84]	8 men, 17.0 ± 1.1	Rugby	12/2	Squats, bench press, 4–6 sets/3–10 reps (60–90%)		10 m: 2.54	10 m: 0.41
Harries, et al. [84]	8 men, 16.8 ± 1.0	Rugby	12/2	Squats, bench press, 4–6 sets/3–10 reps (60–90%)		10 m: 1.58	10 m: 0.89
McGawley, et al. [85]	9 men, 23 ± 5	Football	5/2	Squats, lunges, cleans and other variants 2–3 sets/5–10 reps (75–90%)	Football, Sprint, jumping and agility before strength training, and one session of explosive exercises	10 m: 1.12	10 m: 0.4
McGawley, et al. [85]	9 men, 23 ± 4	Football	5/2	Squats, lunges, cleans and other variants 2–3 sets/5–10 reps (75–90%)	Football, Sprint, jumping and agility before strength training, and one session of explosive exercises	10 m: 2.30	10 m: 0.63
Moir, et al. [86]	10 men, 18.9 ± 1.7	Physically active	8/3	Squats, bench press, push-press, flys, power cleans, deadlifts, shrugs (week 1–4: 3 sets/12 RM, week 5–8: 3 sets/5 RM)		10 m: −5.98 10–20 m: 6.57	10 m: −0.85 10–20 m: 1.13
Pareja-Blanco, et al. [68]	11 men, 23.3 ± 3.2	Train with weights	6/3	Squats, 3–4 sets/2–8 reps (60–80%)		10 m: 1.12	10 m: 0.25
Speirs, et al. [87]	9 men, 18.1 ± 0.5	Rugby	5/2	Single leg squat 4 sets/3–6 reps (75–92%)	Rugby	10 m: 1.73	10 m: 0.41

**Table 5 sports-10-00148-t005:** An overview of the collective mean ± SD of percentage change and effect size for each of the different types of training interventions.

Type of Training Intervention	Mean ± SD Change (%)	Mean ± SD Effect Size
*10-m sprint times*		
Maximal weight training interventions (85–100% of 1-RM)	4.23 ± 3.67	1.48 ± 1.32
Explosive weight training with moderate load interventions (60–85% of 1-RM)	2.78 ± 3.22	0.33 ± 0.97
Explosive weight training with light load interventions (30–60% of 1-RM)	0.02 ± 1.56	0.44 ± 0.35
Hypertrophy training interventions (60–85% of 1RM)	0.47 ± 2.53	0.21 ± 0.53
*Flying 10-m sprint times*		
Maximal weight training interventions (85–100% of 1-RM)	0.91 ± 0.59	0.30 ± 0.27
Explosive weight training with moderate load interventions (60–85% of 1-RM)	0.78 ± 0.21	0.17 ± 0.14
Explosive weight training with light load interventions (30–60% of 1-RM)	1.07 ± 0.98	0.27 ± 0.23
Hypertrophy training interventions (60–85% of 1RM)	6.57 ± 0.00	1.13 ± 0.00

## Data Availability

Not applicable.

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
