# Peer review of "The Effects of Different Relative Loads in Weight Training on Acceleration and Acceleration from Flying Starts"

_sports, 2022, doi:10.3390/sports10100148_

Round 1

Reviewer 1 Report

The topic of the paper is interesting and fits the scope of the journal. The text is relatively well written and composed. I have only minor comments that I believe that help to improve the paper. Congratulations to the authors for this good work

Minor comments

Line 35. Please correct “America football” with “American football”.

Lines 123-124. Link better the two paragraphs.

Author Response

We have made the changes that the reviewer recommended and think that it is now suitable for publication.

The topic of the paper is interesting and fits the scope of the journal. The text is relatively well written and composed. I have only minor comments that I believe that help to improve the paper. Congratulations to the authors for this good work

Minor comments

Line 35. Please correct “America football” with “American football”.

Changed now

Lines 123-124. Link better the two paragraphs.

We have changed it in: Another type of strength training to improve speed is explosive strength training with light loads, also called power training, which …

To link it better to the previous paragraph.

Reviewer 2 Report

The study is well designed according to the requirements of the scientific method and based on solid literature. The article is well-written and the conclusions are interesting and important for sports practice.

Author Response

The study is well designed according to the requirements of the scientific method and based on solid literature. The article is well-written and the conclusions are interesting and important for sports practice.

Thank you for this comment.

Reviewer 3 Report

General Comments to the authors:

The main aim of this review was to analyze the effect of relative load during resistance training on acceleration and acceleration from flying starts. Improving acceleration capacity is a determining factor in many sports modalities. Countless training methods have been postulated to improve this ability, including resistance training. However, the effect of resistance training programs on changes in acceleration capacity may depend on how the training is configured. Therefore, I think that the topic of this paper is appropriate, as there is a need to provide valid data to guide the training of coaches and strength and conditioning professionals. However, I think that the current review presents several methodological limitations preventing adequately address this problem.

Below, I will list some of the issues that I consider to be fundamental and that are wrong in this review:

1.     The title and aims stated are wrong. This study does not analyze "The effects of different approaches (intensities) in strength training on..." but rather, what is intended is to analyze "the effect of relative load during resistance training on...". It might seem the same, but it has nothing to do one thing with another. First, as indicated in the text, “The results were categorized into four groups: 1) explosive strength training with light loads at 30–60% of 1-RM, 2) explosive strength training with moderate loads at 60–85% of 1-RM, 3) maximal strength training at 85–100% of 1-RM, and 4) hypertrophy training at 60–85% of 1-RM.” The main difference between the training programs is the relative load used. Thus, that variable should be the main “independent variable” of this review (in the following points I will clarify more questions on this aspect). Second, the studies included in the current review used resistance training programs. This is a type of strength training, just like many others, such as plyometric training or resisted sprint training. That is, strength training is a general concept, and "resistance training" is a type of training included within "strength training".

2.     The main limitation of this review is that the effect of resistance training depends on a myriad of variables, and not only of the relative load. Variables such as number of sets, repetitions per set, frequency, type and order of the exercises, number of exercises per session, recovery time between repetitions, sets, exercises and sessions, execution velocity in each repetition, etc., they are essential to determine the training stimulus and, as a consequence, the adaptations to training. Therefore, proposing a review to determine the effect of relative load during resistance training on acceleration capacity (or on any other performance variable) is, to say the least, a difficult chimera to resolve. Only a tiny approximation to the problem could be made, but with many limitations, as is the case with this review. Recommendations on "which relative intensity is more beneficial" could only be provided if it were found that the degrees of fatigue (or level of effort) were similar between training programs with different relative loads. Otherwise, any conclusion we can draw would be false. Since this review has "structured" the groups according to the relative load used, without taking into account other variables such as volume, type of exercises, training frequency, etc., the comparisons between groups are mere approximations without foundation.

3.     Another aspect to take into account is that, in this review, different training methods are not compared. The training method is the same: "resistance training". The difference is in how the different variables that "configure" the resistance training program are "manipulated". This is a serious misconception that manifests itself throughout the document. Similarly, they could not be considered as "different approaches" either.

4.     Another limiting aspect of this review is the terminology used to determine the different “types of training”. "Explosive training", "hypertrophy training", "power training" is continually mentioned throughout the document as if they were different trainings. The main goal of RT is to increase movement velocity against a given absolute load, it could be considered that the only possible type of RT is training for improving maximal strength, since to improve the movement velocity it is necessary to improve the applied force to a given absolute load. Thus, one of the most common mistakes or misconceptions is to believe that power, ballistic, or velocity training are different types of training than maximal strength training, or to consider that the effects of these alleged types of RT occur without improving the maximal (peak) force applied to the loads used to assess performance. This may seem somewhat minor, but it is a fundamental issue of this review, since different training programs are proposed through terminology, but they are not defined through "number", which is what is really important and relevant to know the possible training effect. These and other misconceptions manifest themselves throughout the Introduction (thus weakening the justification for the review) and the Discussion sections (making the rationale seem unsound).   

Therefore, in my opinion, this manuscript does not fulfill the minimum requirements for publication in this journal.

Author Response

We have made the changes that the reviewer recommended and think that it is now suitable for publication. We have colored all changes in the manuscript red.

  1. The title and aims stated are wrong. This study does not analyze "The effects of different approaches (intensities) in strength training on..." but rather, what is intended is to analyze "the effect of relative load during resistance training on...". It might seem the same, but it has nothing to do one thing with another. First, as indicated in the text, “The results were categorized into four groups: 1) explosive strength training with light loads at 30–60% of 1-RM, 2) explosive strength training with moderate loads at 60–85% of 1-RM, 3) maximal strength training at 85–100% of 1-RM, and 4) hypertrophy training at 60–85% of 1-RM.” The main difference between the training programs is the relative load used. Thus, that variable should be the main “independent variable” of this review (in the following points I will clarify more questions on this aspect). Second, the studies included in the current review used resistance training programs. This is a type of strength training, just like many others, such as plyometric training or resisted sprint training. That is, strength training is a general concept, and "resistance training" is a type of training included within "strength training".

We have changed the title and the manuscript and used relative loads and weight training to cover the different training programs. We think that this will be better for the readers to understand. Since we did only involve weight training we think it is better to use weight training than resistance training since we did not include resisted sprint or plyometric training which are also resistance training. In the literature weight, resistance and strength training are used synonymous to each other. We think that the reader easy could identify weight training with strength training as it is both resistance training. The whole purpose of the weight training in the present review is to show what the purpose was of the weight training – maximal strength, explosive strength etc and what it would mean for sprint performance when using this type of training. We hope the reviewer agrees with our point of view. We have now made a distinction about weight training interventions, and we speak about strength wen it is about strength gains. In addition, it is all resistance training with weights. Thank you for the comment.

  1. The main limitation of this review is that the effect of resistance training depends on a myriad of variables, and not only of the relative load. Variables such as number of sets, repetitions per set, frequency, type and order of the exercises, number of exercises per session, recovery time between repetitions, sets, exercises and sessions, execution velocity in each repetition, etc., they are essential to determine the training stimulus and, as a consequence, the adaptations to training. Therefore, proposing a review to determine the effect of relative load during resistance training on acceleration capacity (or on any other performance variable) is, to say the least, a difficult chimera to resolve. Only a tiny approximation to the problem could be made, but with many limitations, as is the case with this review. Recommendations on "which relative intensity is more beneficial" could only be provided if it were found that the degrees of fatigue (or level of effort) were similar between training programs with different relative loads. Otherwise, any conclusion we can draw would be false. Since this review has "structured" the groups according to the relative load used, without taking into account other variables such as volume, type of exercises, training frequency, etc., the comparisons between groups are mere approximations without foundation.

We fully agree that volume, sets, repetition, exercises etc are also important and which makes it difficult to compare the different studies with each other. However, in the training world these 4 different types of weight training are used to enhance sprint performance and the review would at least give a good overview over what is done and how it enhances sprint performance or not. The reader can self-decide if they are going to use of training program with the volume etc. We think, it is good for the reader to get an overview and not just a very limited view over one or few studies they have access to. To discuss in detail all the different volumes between the studies would be too detailed for this review.

  1. Another aspect to take into account is that, in this review, different training methods are not compared. The training method is the same: "resistance training". The difference is in how the different variables that "configure" the resistance training program are "manipulated". This is a serious misconception that manifests itself throughout the document. Similarly, they could not be considered as "different approaches" either.

We have changed it in programs or interventions to avoid confusion about different methods since it all are the same training method “weight training” or also called “resistance training”. We have changed it in the whole manuscript according to the comment of the reviewer.

  1. Another limiting aspect of this review is the terminology used to determine the different “types of training”. "Explosive training", "hypertrophy training", "power training" is continually mentioned throughout the document as if they were different trainings. The main goal of RT is to increase movement velocity against a given absolute load, it could be considered that the only possible type of RT is training for improving maximal strength, since to improve the movement velocity it is necessary to improve the applied force to a given absolute load. Thus, one of the most common mistakes or misconceptions is to believe that power, ballistic, or velocity training are different types of training than maximal strength training, or to consider that the effects of these alleged types of RT occur without improving the maximal (peak) force applied to the loads used to assess performance. This may seem somewhat minor, but it is a fundamental issue of this review, since different training programs are proposed through terminology, but they are not defined through "number", which is what is really important and relevant to know the possible training effect. These and other misconceptions manifest themselves throughout the Introduction (thus weakening the justification for the review) and the Discussion sections (making the rationale seem unsound).

We agree that all the type of training is resistance training. However, in the literature it is mostly referred to these types of training. We have changed it now in the entire manuscript in weight training with the different focus they have (low loads, moderate loads, maximal loads etc). Most trainers and researchers still use the terms we have used. We hope that with the changes we have made it is not any misconception. The other two reviewers did not have any comments on these terms, which also could be an indication that more reviewers have the same understanding about the terms used in the review.     

Round 2

Reviewer 3 Report

General Comments to the authors:

I thank and acknowledge the great work done by the authors to improve the manuscript. However, I think that the essential errors that were in the original document remain in this revised version.

With respect to the first and second comment, I think that the authors have not perceived the depth of the comments that I made in the previous review. It is not enough to change "strength training" to "resistance training or weight training". The problem is that the authors intend to "decode" the importance of a variable (relative load) without controlling the rest of the variables influencing the process of training adaptation. In this regard, within the same group of relative loads (e.g., Maximal weight training interventions) we can find training groups with different volume, types of exercises, number of exercises, different sample (men or women), or different performance of the subjects (soccer, rugby, rowing, basketball, etc.). It is obvious that all these factors have a high influence on the training effect and have not been considered in the present review.

The authors' argument is that “in the training world these 4 different types of weight training are used to enhance sprint performance and the review would at least give a good overview over what is done and how it enhances sprint performance or not. The reader can self-decide if they are going to use of training program with the volume etc. We think, it is good for the reader to get an overview and not just a very limited view over one or few studies they have access to. To discuss in detail all the different volumes between the studies would be too detailed for this review. However, contrary to what the authors state, I think that a review article of these characteristics is what can make readers have a "reductionist view" of resistance training. For readers, the only thing they are going to "get" from this review is that "maximal weight training is the best method to improve acceleration", when that is nothing more than a mathematical artifice without any foundation.

With regard to the last of my concerns, the authors indicate in their responses that “in the training world these 4 different types of weight training are used to enhance sprint performance”, “in the literature it is mostly referred to these types of training”, “Most trainers and researchers still use the terms we have used” and “The other two reviewers did not have any comments on these terms, which also could be an indication that more reviewers have the same understanding about the terms used in the review.”. Indicating that one thing is done because "others" also do it, seems to me an argument that is too weak in the field of Science. Perhaps it is time to start changing all those things that are done wrong in our research area and not base or justify our research on erroneous and sometimes "absurd" concepts. A very clear example of what I want to express is the so-called explosive training. Explosive means "that makes or can make an explosion". And my question is, is there something that explodes or can explode in the human body? The answer, obviously, is no. So, why is one type of training called "explosive"? I hope you do not receive these comments in a negative way, I just try to put some sanity within my sphere of responsibility

Therefore, in my opinion, as comment in the previous review, this manuscript does not fulfill the minimum requirements for publication in this journal. However, this final decision rests with the Editor-in-Chief, for which I am not responsible and have virtually no influence.

Author Response

With respect to the first and second comment, I think that the authors have not perceived the depth of the comments that I made in the previous review. It is not enough to change "strength training" to "resistance training or weight training". The problem is that the authors intend to "decode" the importance of a variable (relative load) without controlling the rest of the variables influencing the process of training adaptation. In this regard, within the same group of relative loads (e.g., Maximal weight training interventions) we can find training groups with different volume, types of exercises, number of exercises, different sample (men or women), or different performance of the subjects (soccer, rugby, rowing, basketball, etc.). It is obvious that all these factors have a high influence on the training effect and have not been considered in the present review.

As said before we fully agree that all these parameters can have an influence upon the sprint performance. We have now included this to the text as limitations of the reviewer. The make the reader aware of this shortcoming of the present study. However, we think that it is difficult to calculate the total training volume and discuss the effect of the other factors in the present review since this is almost not possible. However, we still think that the categorization we have made can give some more insight for the reader about the effect of using relative loads in weight training upon sprint performance, which is very normal to use in sports training.

We have included this to the text:

It has to be noticed that the present revew has some limitiations. Firstly, the different weight training interventions were categorised after different relative loads and did not investigate the effect of the different and number of weight exercises used (squats, hip trusts, etc.). Secondly, the total work load (weeks of training, number of repetitions and sets, etc.) of each intervention was not discussed in the review, which can have a large effect upon the findings related to the sprint performance. Thirdly, different samples (men and women) from different sports with different performance were investigated. All these extra parameters make it very difficult to calculate the total workload of each intervention and the effect of this on sprint in performance in different samples (soccer, rugby, rowing, basketball, etc.). Therefore, the reader must be aware of these shortcomings when using the findings of the present review.

The authors' argument is that “in the training world these 4 different types of weight training are used to enhance sprint performance and the review would at least give a good overview over what is done and how it enhances sprint performance or not. The reader can self-decide if they are going to use of training program with the volume etc. We think, it is good for the reader to get an overview and not just a very limited view over one or few studies they have access to. To discuss in detail all the different volumes between the studies would be too detailed for this review. However, contrary to what the authors state, I think that a review article of these characteristics is what can make readers have a "reductionist view" of resistance training. For readers, the only thing they are going to "get" from this review is that "maximal weight training is the best method to improve acceleration", when that is nothing more than a mathematical artifice without any foundation.

Here we have another point of view than the reviewer. We think that the review still gives new insights over the use of different weight training interventions based upon relative loads upon sprint performance.

With regard to the last of my concerns, the authors indicate in their responses that “in the training world these 4 different types of weight training are used to enhance sprint performance”, “in the literature it is mostly referred to these types of training”, “Most trainers and researchers still use the terms we have used” and “The other two reviewers did not have any comments on these terms, which also could be an indication that more reviewers have the same understanding about the terms used in the review.”. Indicating that one thing is done because "others" also do it, seems to me an argument that is too weak in the field of Science. Perhaps it is time to start changing all those things that are done wrong in our research area and not base or justify our research on erroneous and sometimes "absurd" concepts. A very clear example of what I want to express is the so-called explosive training. Explosive means "that makes or can make an explosion". And my question is, is there something that explodes or can explode in the human body? The answer, obviously, is no. So, why is one type of training called "explosive"? I hope you do not receive these comments in a negative way, I just try to put some sanity within my sphere of responsibility

Here we differ with the reviewer. We only use categories that are used regularly in training and sports science literature. We have not invented these explosive weight training. That is from the previous literature. We agree with the reviewer that explosive weight training is perhaps not a good expression. However, as said before we only use terminology that is used in strength training on a daily basis and is educated at many universities etc. Perhaps the reviewer could come with another terminology on this matter?